# A non-canonical mechanism of GPCR activation

Alexander S. Powers[1,2,3,4,5,15], Aasma Khan[6,11,15], Joseph M. Paggi[2,3,4,5], Naomi R. Latorraca[2,3,4,5,7,12], Sarah Souza [6], Jerry Di Salvo[8], Jun Lu[9,13], Stephen M. Soisson [9,14], Jennifer M. Johnston[10], Adam B. Weinglass[6] & Ron O. Dror [2,3,4,5,7] ✉

The goal of designing safer, more effective drugs has led to tremendous interest in molecular mechanisms through which ligands can precisely manipulate the signaling of G-protein-coupled receptors (GPCRs), the largest class of drug targets. Decades of research have led to the widely accepted view that all agonists—ligands that trigger GPCR activation—function by causing rearrangement of the GPCR's transmembrane helices, opening an intracellular pocket for binding of transducer proteins. Here we demonstrate that certain agonists instead trigger activation of free fatty acid receptor 1 by directly rearranging an intracellular loop that interacts with transducers. We validate the predictions of our atomic-level simulations by targeted mutagenesis; specific mutations that disrupt interactions with the intracellular loop convert these agonists into inverse agonists. Further analysis suggests that allosteric ligands could regulate the signaling of many other GPCRs via a similar mechanism, offering rich possibilities for precise control of pharmaceutically important targets.

One-third of existing drugs act by binding to G-protein-coupled receptors (GPCRs), and these receptors also represent the largest class of targets for the development of new therapeutics[1,2]. A tremendous amount of work has focused on understanding the molecular mechanism of GPCR activation—that is, how drugs and naturally occurring ligands cause GPCRs to adopt molecular conformations that stimulate intracellular signaling.

For decades, the dominant model of GPCR activation has been that agonists (i.e., ligands that trigger receptor signaling) work by facilitating rearrangement of a GPCR's seven conserved transmembrane (TM) helices[1,3–8]. These TM helices connect the extracellular and intracellular surfaces of the GPCR, allowing extracellular ligands to cause the opening of a large intracellular pocket in which G proteins and other intracellular signaling proteins bind. Molecular structures, spectroscopic experiments, mutagenesis studies, and computer simulations involving many GPCRs and ligands have all supported this model, leading to the common assumption that all GPCR agonists act by facilitating rearrangement of the TM helices, which then stabilizes G protein binding[9–13].

[1]Department of Chemistry, Stanford University, Stanford, CA, USA. [2]Department of Computer Science, Stanford University, Stanford, CA, USA. [3]Department of Molecular and Cellular Physiology, Stanford University School of Medicine, Stanford, CA, USA. [4]Department of Structural Biology, Stanford University School of Medicine, Stanford, CA, USA. [5]Institute for Computational and Mathematical Engineering, Stanford University, Stanford, CA, USA. [6]Department of Quantitative Biosciences, Merck & Co., Inc., Rahway, NJ, USA. [7]Biophysics Program, Stanford University, Stanford, CA, USA. [8]Evotec, Princeton, NJ, USA. [9]Department of Structural Chemistry, Merck & Co., Inc., West Point, PA, USA. [10]Department of Modeling and Informatics, Merck & Co., Inc., Rahway, NJ, USA. [11]Present address: Department of Therapeutic Proteins, Regeneron Pharmaceuticals Inc., Tarrytown, NY, USA. [12]Present address: Department of Biochemistry and Molecular Biophysics, Columbia University Irving Medical Center, New York, NY, USA. [13]Present address: Small Molecule Discovery, Zai Lab (US) LLC, Cambridge, MA, USA. [14]Present address: Protein Therapeutics and Structural Biology, Odyssey Therapeutics, Boston, MA, USA. [15]These authors contributed equally: Alexander S. Powers, Aasma Khan. ✉e-mail: ron.dror@stanford.edu

Most known GPCR ligands bind at the orthosteric site where endogenous ligands bind, but a recent explosion of GPCR structures has demonstrated that various ligands can bind to diverse sites across the GPCR surface[2,14–19]. Ligands that bind in such allosteric sites are of great interest for drug discovery, not least because they provide a mechanism to achieve high selectivity for target receptors[14,20]. Many allosteric ligands modulate the effects of endogenous orthosteric ligands, with positive allosteric modulators increasing these effects and negative allosteric modulators reducing them[21–24]. A few allosteric ligands also act as agonists, capable of stimulating receptor activation and signaling on their own[16,25–27]. Like orthosteric ligands, allosteric ligands—including those that bind near the intracellular surface—are widely assumed to act by causing or preventing the rearrangement of a GPCR's transmembrane helices. Studies of multiple allosteric ligands have supported this model[22–24,26,28].

In this work, we investigate the detailed molecular mechanism by which AP8, an allosteric ligand that binds near the intracellular side of free fatty acid receptor 1 (FFAR1 or GPR40)[16,17], activates this receptor (Fig. 1a). AP8 is a full agonist—it strongly stimulates activation of FFAR1 even in the absence of other ligands. To our surprise, we discover that AP8 acts via a mechanism fundamentally different from that of previously characterized allosteric and orthosteric GPCR ligands. Instead of favoring rearrangement of the TM helices before the G protein binds, AP8 changes the orientation of an intracellular loop, leading the

receptor to couple more effectively to G proteins—with the TM helices rearranging only as a result of G protein binding. Our results suggest that several other FFAR1 agonists also act via this mechanism and that ligands could control signaling of many other GPCRs in a similar fashion.

These findings indicate that the classical model of GPCR activation is incomplete: ligands can trigger activation not only by causing the opening of the transducer-binding pocket but also by directly rearranging intracellular receptor loops—with different agonists capable of activating the same receptor by either mechanism. Our results suggest a variety of opportunities for designing drugs that precisely target various GPCRs and provide fine control over their signaling.

## Results

### Agonist AP8 does not control transmembrane helix conformation

Previous hypotheses for the mechanism by which AP8 activates FFAR1 have been based on a comparison of two crystal structures: one of FFAR1 bound to both AP8 and the orthosteric partial agonist MK-8666 (AP8-bound structure), and one of FFAR1 bound only to MK-8666 (AP8-free structure). These structures differ primarily in two ways (Supplementary Figs. 1, 2)[16]. First, TM5 is shifted 3 Å toward the extracellular end of the receptor relative to TM4 in the AP8-bound structure, along with small shifts in the intracellular ends of TM3 and

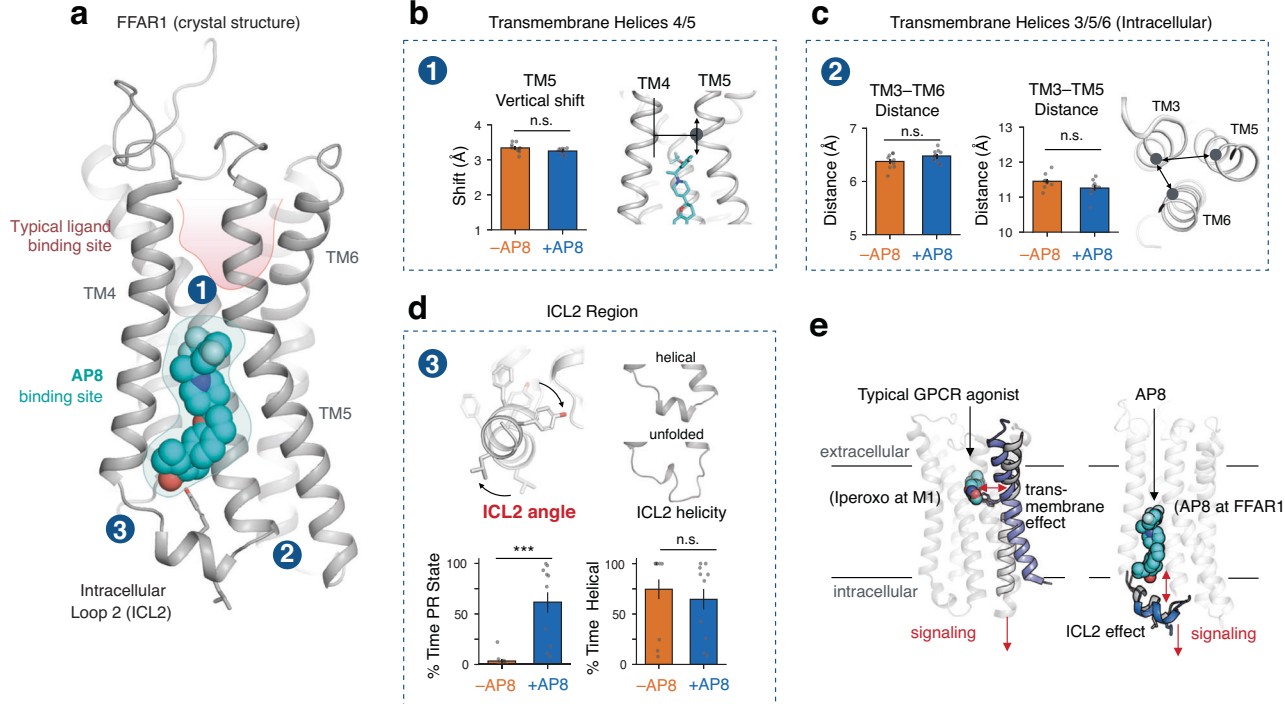

**Fig. 1 | Unlike typical GPCR ligands, AP8 controls the conformation of an intracellular loop but does not affect key transmembrane helices. a** The crystal structure of FFAR1 (PDB 5TZY) shows agonist AP8 (cyan) bound to a membrane-facing pocket near intracellular loop 2 (ICL2) and transmembrane helices 3, 4, and 5 (TM3, TM4, and TM5). Other FFAR1 agonists, such as MK-8666, bind in a more typical pocket (highlighted in red) within the extracellular region of the receptor. **b, c** AP8 does not significantly affect the conformation of key TM helices in molecular dynamics simulations. The conformation of TM5 was measured by its vertical shift relative to TM4 (see Methods), which differs by 3 Å between the AP8-bound and AP8-free crystal structures but converges to the same conformation in simulations ($P = 0.11$, two-sided Mann-Whitney U test, $N = 10$ independent simulations). The intracellular conformation of TM helices was measured by distances between Cα atoms ($P = 0.10$ TM3-TM6, residue 105 Cα to 222 Cα, $P = 0.85$ TM3-TM5, 104 Cα to 208 Cα, two-sided MWU test, N = 10 independent simulations). Data

presented as mean with 68% confidence interval (CI). Blue bars are simulations started from AP8-bound crystal structure and orange bars are simulations started from AP8-free crystal structure. **d** AP8 has a significant effect on the orientation of the ICL2 helix in simulation as measured by the rotation about the helical axis (see Methods) ($P = 0.0004$, two-sided MWU test, $N = 10$ independent simulations). The PR state is the "positively rotated" state of the ICL2 helix observed in the AP8-bound crystal structure. AP8 does not have a significant effect on how frequently ICL2 adopts a helical conformation ($P = 0.22$, two-sided MWU test, $N = 10$ independent simulations). The conformation of ICL2, both angle and helicity, was quantified over simulations with and without AP8 bound, starting from the AP8-bound crystal structure. Data presented as mean with 68% CI. **e** Diagram showing the proposed mechanism for AP8's agonism in comparison to a canonical GPCR agonist (iperoxo at M1). AP8 directly alters the receptor's intracellular surface without long-range rearrangements of transmembrane helices.

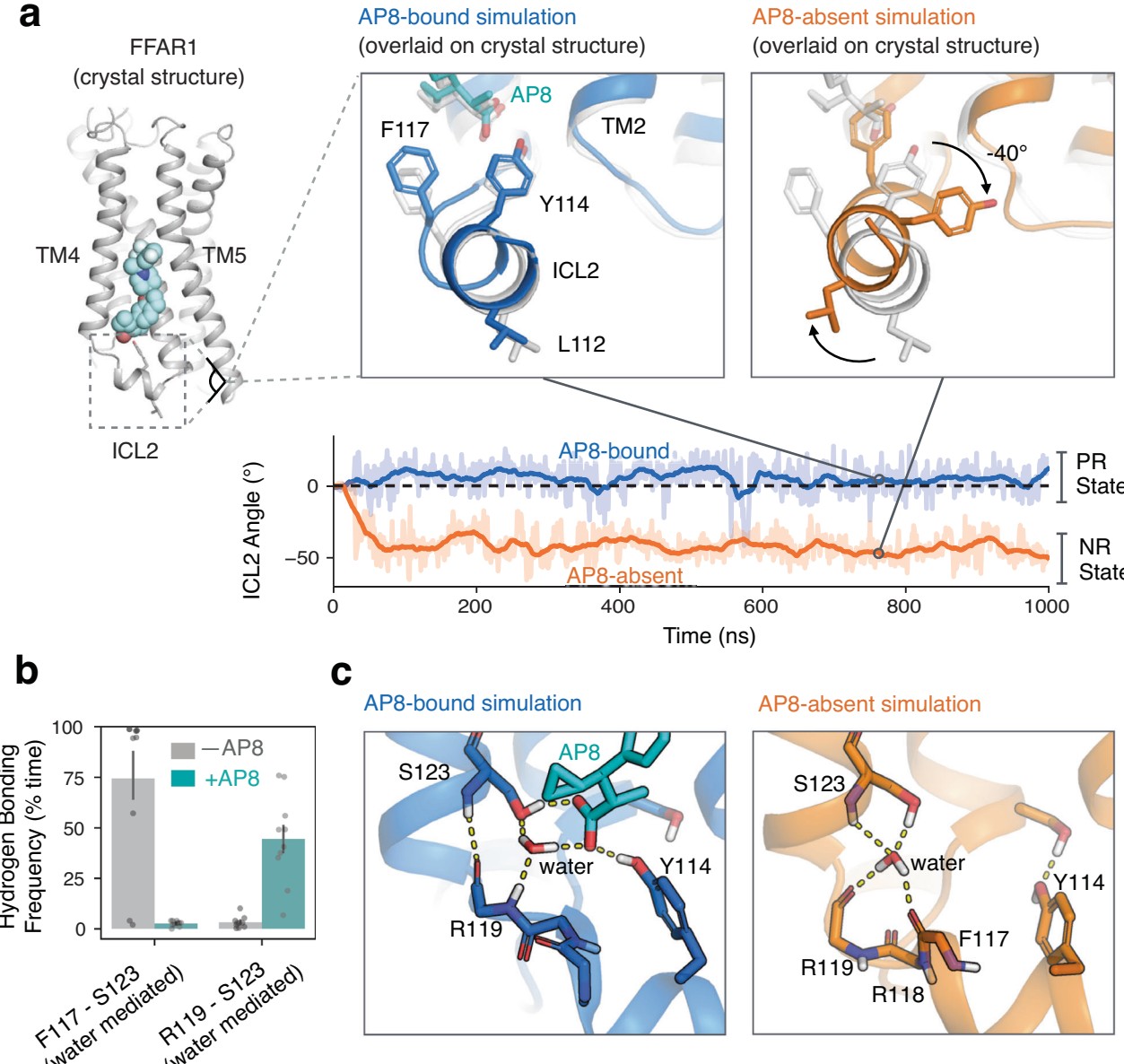

**Fig. 2 | Ligand directly affects the intracellular receptor surface by rotating intracellular loop 2. a** Simulations of the receptor with AP8 bound (blue) favor one stable ICL2 conformation (PR state) while simulations without AP8 (orange) favor a distinct negatively rotated ICL2 conformation (NR state). Both simulations were started from the same structure. Representative simulation snapshots are shown at indicated time points overlaid on the AP8-bound crystal structure (light grey). Simulation trajectories show the ICL2 angle as measured by the rotation of L112 and Y114 around the ICL2 helix axis (see Methods). The dashed horizontal line at 0 degrees is the distance in the starting structure. **b** AP8 alters a network of polar interactions; the frequency of water-mediated hydrogen bonds between key ICL2 residues was quantified in the presence and absence of AP8. Data presented as mean with 68% CI ($N = 10$ independent simulations for each condition). Only simulation frames where ICL2 remained folded were used. **c** Simulation frames show representative hydrogen bonds (yellow dashes) formed with and without AP8 bound. With AP8 bound, a stable water molecule forms a hydrogen bond network bridging AP8 and the ICL2 backbone. In the absence of AP8, the water molecule reorients to form a new stable network of hydrogen bonds, necessitating a rotation of ICL2.

TM6. Second, intracellular loop 2 (ICL2) is helical in the AP8-bound structure but unresolved in the AP8-free structure. These differences led to two hypotheses for the mechanism of agonism for AP8 and related ligands[16]: (1) these ligands may stabilize key transmembrane helices including TM5 and TM6 in the canonically "active" conformation that enables G protein binding, or (2) these ligands promote a helical ICL2 over a disordered ICL2 to directly stabilize part of the interface for G protein binding.

To investigate these hypotheses, we performed extensive molecular dynamics (MD) simulations of FFAR1 in a hydrated lipid bilayer, with and without AP8 bound. We initiated simulations from the AP8-bound structure; we retained AP8 under one condition and

removed it in the other. We also initiated simulations from the AP8-free structure.

Strikingly, we observed that AP8 had little influence on the arrangement of transmembrane helices in these simulations (Fig. 1b, c, Supplementary Fig. 1a, b). In 2-µs simulations initiated from the AP8-bound structure, the removal of AP8 had minimal effect on the distances between TM helices. Moreover, in simulations initiated from the AP8-free structure, the TM helices quickly—within 200 ns—adopted positions seen in the AP8-bound structure (Supplementary Fig 2c, Supplementary Fig. 3). In other words, the differences between the AP8-bound and AP8-free crystal structures, including the 3 Å difference in TM5 position, did not persist in simulations and did not depend

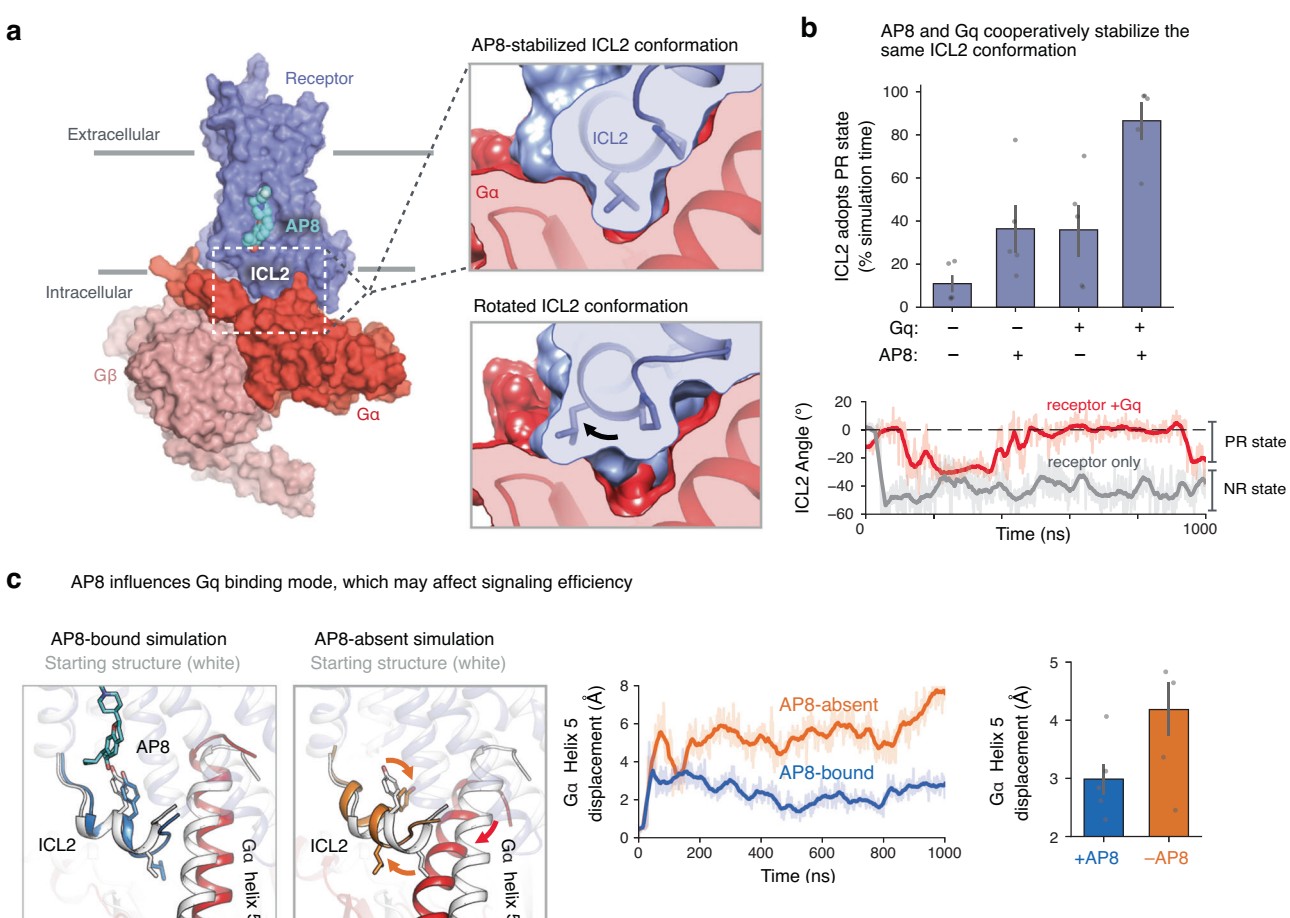

**Fig. 3 | ICL2 helix orientation affects G protein binding. a** Model of active FFAR1 with bound AP8 and heterotrimeric Gq constructed from homology modeling and alignment with other complexes (see Methods). Zoomed image shows that ICL2 in the AP8-stabilized conformation (PR state) forms a tight interface with Gqα (red). In the lower image, ICL2 modeled in the NR state has poor shape complementarity with the Gq surface. **b** AP8 and Gq both independently stabilize the PR state of ICL2, and do so to an even greater degree together, indicating a cooperative effect. Data presented as mean with 68% CI (N = 5 independent simulations for each condition). The simulation trace below shows ICL2 angle vs. time for the receptor-only condition (grey) and receptor-Gq condition (red). When Gq is bound to the receptor, the ICL2 PR state is stabilized relative to the receptor alone. The dashed horizontal line at 0 degrees is the distance in the AP8-bound crystal structure. **c** ICL2 conformation is also coupled to the orientation of Gq relative to the receptor, in particular Gα helix 5. Representative simulation frames (left) and traces (middle) of the FFAR1-Gq model are shown with and without AP8 bound. In images, the starting structure is shown in grey, and the simulation frame in color. The displacement of the Gα helix 5 was calculated by aligning simulation frames on the receptor and calculating the root mean square displacement (RMSD) of helix 5 (terminal 10 residues) relative to the starting structure. Bars (right) show mean displacement with 68% CI (N = 5 independent simulations for each condition).

on the presence or absence of AP8. We found that the difference in TM3, TM5 and TM6 position between the crystals may be explained by a difference in crystal packing contacts around these helices (Supplementary Fig. 2b). In additional control simulations without MK-8666 present, the removal of AP8 again had little effect on the distances between TM helices (Supplementary Fig. 1b). By contrast, in simulations of FFAR1 with and without the orthosteric agonist MK-8666 bound, we observed substantial differences in the arrangement of transmembrane helices (Supplementary Fig. 1c). These motions are typical of those caused by GPCR agonists and provide a reference point for the atypical behavior of AP8[3,4].

The AP8-bound and AP8-free crystal structures of FFAR1 represent inactive receptor states—the intracellular cavity is closed in these structures, such that it cannot fit a G protein. The timescales of our simulation are not long enough to sample transitions between the inactive and active states of the receptor. Therefore, we also examined interactions between AP8 and the transmembrane helices starting from the active receptor state, with a G protein bound, to investigate potential conformational changes around the AP8 binding site that might be linked to receptor activation. A recently published cryo-EM

structure of the active-state FFAR1-Gq complex[29] shows that the TM helix residues immediately surrounding the AP8 binding site undergo very little conformational change upon receptor activation (Supplementary Fig. 4a). This is in contrast to the TM helix residues surrounding the orthosteric binding site, which do undergo significant conformational changes between the active and inactive states (Supplementary Fig. 5). Additionally, using simulations of the FFAR1-Gq-AP8 complex (see Methods), we calculated interaction energies and distances between AP8 and surrounding TM helix residues (Supplementary Fig. 4b) in both active and inactive receptor states across simulation frames. There were no favorable changes in these interactions between the inactive and active states, again supporting that AP8 does not act by directly stabilizing the active-state conformation of the TM helices.

As part of the process by which FFAR1 binds to a G protein, transmembrane helix 6 (TM6) of FFAR1 would necessarily undergo the canonical outward movement to accommodate the binding of the G protein. Our observation that the presence of AP8 has little influence on the arrangement of the FFAR1 TM helices in simulations suggests that AP8 may cause this TM6 movement only indirectly, by increasing

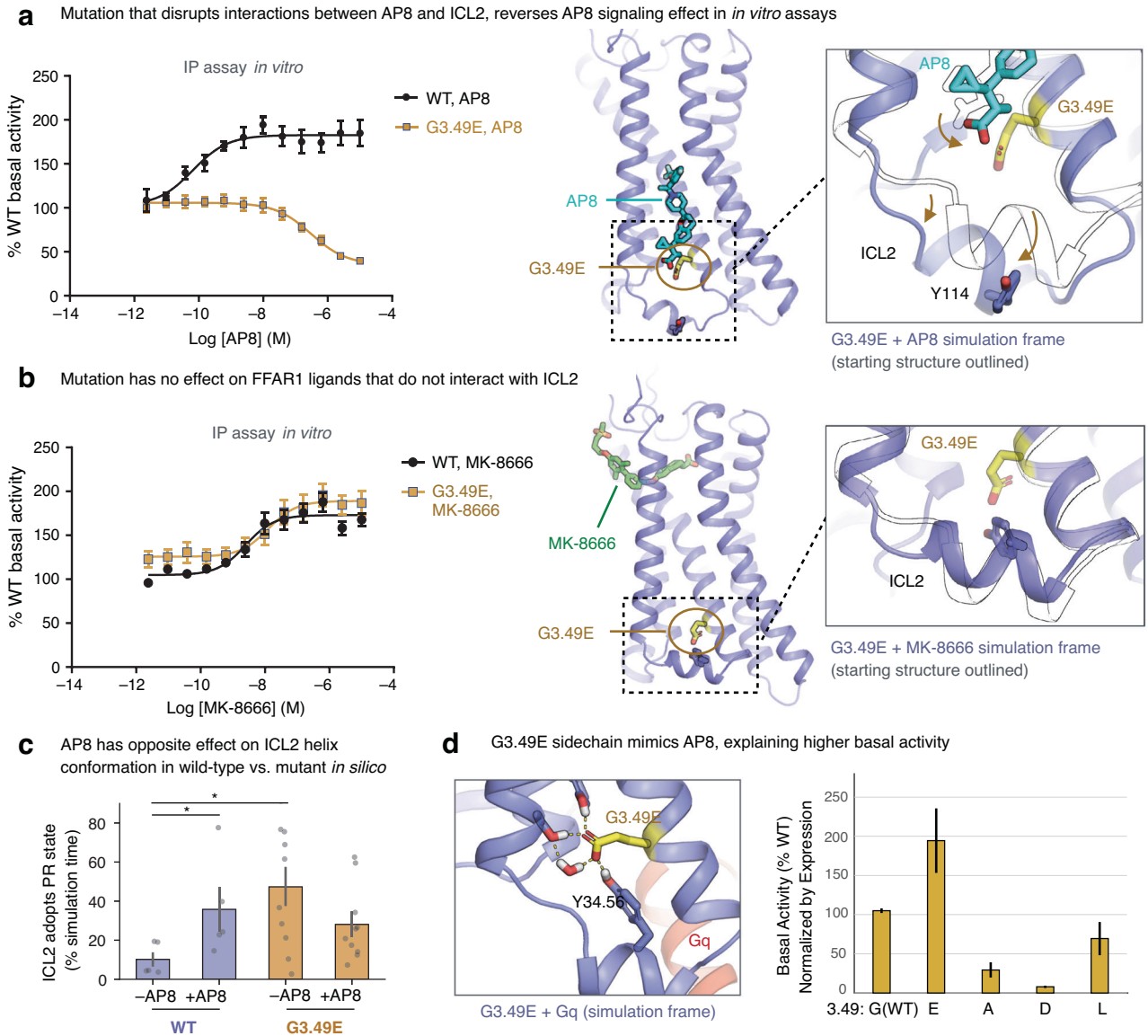

**a** Mutation that disrupts interactions between AP8 and ICL2, reverses AP8 signaling effect in *in vitro* assays

**b** Mutation has no effect on FFAR1 ligands that do not interact with ICL2

**c** AP8 has opposite effect on ICL2 helix conformation in wild-type vs. mutant *in silico*

**d** G3.49E sidechain mimics AP8, explaining higher basal activity

**Fig. 4 | Mutagenesis experiments validate computationally derived activation mechanism.** At the mutated G3.49E receptor, AP8 acts as an inverse agonist. FFAR1 activity was monitored in IP1 accumulation assays in HEK293 cells expressing WT or G3.49E mutant receptors treated with **a** AP8 or **b** MK-8666. Data is plotted as the % of WT receptor basal activity (cells treated with 1% DMSO), where data points are mean ± S.E.M. from $N = 3$ biologically independent experiments and $n = 2$ technical replicates. Dose-response curves were fit to a standard 4-parameter non-linear regression model. Images at right show simulation frames in color and starting structure as a black outline; AP8 is displaced from its WT binding pose by G3.49E, likely due to the repulsion of the two nearby carboxylates. **c** In simulations, AP8 destabilizes the PR ICL2 state at the G3.49E receptor, the opposite of its behavior at the WT receptor. Data presented as mean with 68% CI ($P = 0.031$ WT −AP8 vs. +AP8, $P = 0.027$ WT −AP8 vs. G3.49E −AP8, two-sided MWU test, from left $N = 5, 5, 10, 10$ independent simulations for each condition). **d** Basal receptor activity in the IP1 accumulation assay, normalized by receptor surface expression, is plotted at right for different mutants at the 3.49 position. Only G3.49E leads to an increase in basal receptor activity relative to WT. At left, a snapshot of the simulation of the G3.49E receptor in complex with Gq shows the glutamate sidechain can mimic the interactions of the AP8 carboxylate. Data are expressed as the mean ± S.E.M. from a single fit to grouped data from N biologically independent experiments (from left, $N = 3, 3, 2, 3, 2$) and $n = 2$ technical replicates.

the likelihood of G protein binding through other mechanisms, rather than causing the TM6 movement directly (i.e., rather than favoring TM6 movement even in the absence of a G protein).

**Allosteric agonist controls intracellular helix orientation**

Next, we examined the alternative hypothesis that AP8 acts as an agonist by stabilizing ICL2 in a helical conformation. Our simulations did not support this hypothesis. First, in simulations initiated from the AP8-bound receptor structure, the ICL2 helix actually unfolds *faster* with AP8 bound than without (Fig. 1d, Supplementary Fig. 6). Second, using adaptively biased MD, we calculated the free energy difference

between helical and disordered ICL2 conformations; the relative stability of the helical conformation did not differ significantly with and without AP8 bound (Supplementary Fig. 7)[30]. Moreover, in several recent experimentally determined structures of GPCR-$G_q$ complexes, ICL2 does not adopt a helical conformation, suggesting that a helical ICL2 conformation is not a requirement for $G_q$ coupling[31,32].

However, AP8 does cause one notable and robust conformational change in FFAR1 in simulation: AP8 controls the equilibrium between two distinct helical ICL2 conformations (Figs. 1d, 2a). In simulations initiated from the AP8-bound structure but with AP8 removed, the ICL2 helix rotates −40 degrees about its helical axis to a distinct

conformation within nanoseconds (Fig. 2a, Supplementary Fig. 6). This rotation hardly ever occurs with AP8 bound. We refer to the helical conformation in the AP8-bound structure as the positively rotated (PR) state, and to the helical conformation adopted upon removal of AP8 as the negatively rotated (NR) state. In additional simulations initiated with ICL2 in the NR state, the ICL2 helix rotated back to the PR state upon the addition of AP8 (Supplementary Fig. 8). These experiments support that AP8 controls the equilibrium between two well-defined ICL2 conformations. Additional control simulations with the orthosteric agonist MK-8666 removed and with engineered mutations reversed showed consistent results (Supplementary Fig. 9a). In the AP8-absent crystal structure, ICL2 is unresolved rather than in a well-defined NR state, but this appears to be explained by crystallization conditions and a bound lipid (Supplementary Fig. 10). In an FFAR1 crystal structure with only an AP8-like allosteric agonist bound, ICL2 adopts the PR state, confirming that the effect of the allosteric agonist on ICL2 is not dependent on the presence of an orthosteric ligand (Supplementary Fig. 11).

How does AP8 trigger this rotation of the ICL2 helix? Simulations reveal that AP8 stabilizes a network of key polar interactions involving ICL2 (Fig. 2b, c). First, a stable water molecule forms hydrogen bonds with both the AP8 carboxylate and the ICL2 backbone. This water is not modeled in the crystal structure, but there is consistent density at this location. In the absence of AP8, this water flips to form a different set of stable hydrogen bonds (Fig. 2c). Second, AP8 forms a frequent hydrogen bond with the sidechain of tyrosine 114 in the middle of the ICL2 helix. To test whether these hydrogen bonds explained the effect of AP8, we purposely disrupted these interactions in additional simulations by mutation or protonation (see Methods). These targeted disruptions prevented AP8's effect, as ICL2 adopted the NR state even with AP8 bound (Supplementary Fig. 9b). We thus concluded that AP8's effect on the orientation of the ICL2 helix is due to the ligand's ability to control this hydrogen bond network. We used this model to design the in vitro mutagenesis experiments discussed below.

### Intracellular helix orientation is coupled to G protein binding

Could a simple rotation of the ICL2 helix control G protein signaling? Structural analysis indicates that ICL2 fits neatly into a corresponding cavity on Gq when the ICL2 helix adopts the AP8-stabilized PR state, but not when the ICL2 helix adopts the NR state (Fig. 3a). This cavity is formed by both the α5 helix and the αN-β1 hinge region of Gq, allowing extensive contacts between the G protein and receptor. Simulations of the full complex with and without AP8 bound further supported this analysis. We observed that the binding of Gq alone—like binding of AP8—favors the PR ICL2 state (Fig. 3b). AP8 and Gq together stabilize this ICL2 conformation to an even greater degree. This implies that AP8 and Gq bind cooperatively to FFAR1: binding of AP8 increases the affinity of FFAR1 for Gq by causing ICL2 to adopt the PR state. We note that as a side-effect of stabilizing the receptor–G protein complex, AP8 would indirectly affect the conformation of TM6. Indeed, rearrangement of the transmembrane helices is still necessary to form this complex, as seen in the FFAR1-Gq cryo-EM structure (Supplementary Fig. 4a). By enhancing the overall stability of the receptor–G protein complex, AP8 effectively promotes the essential interactions between the Gq α5-helix and the receptor core, as well as the interactions between ICL2 and the αN-β1 hinge region, that are important for G protein activation. We note that the contacts between ICL2 and Gq in our active-state simulations are very similar to those in the recent active-state cryo-EM structure (Supplementary Fig. 12).

In addition to modulating the binding affinity of FFAR1 for the G protein, rotation of the ICL2 helix might influence the signaling properties of the FFAR1-G protein complex[33]. Previous studies have indicated that, in other GPCRs, certain ICL2 mutations prevent G protein activation (i.e., GDP-GTP nucleotide exchange) without preventing G protein binding to the receptor[34,35]. This suggests that the ICL2 interface is important for the exchange of GDP for GTP within the G protein. Indeed, when comparing simulations of the FFAR1-Gq complex with and without AP8 bound, we found that the presence of AP8 leads to a change in the orientation of the Gq α5 helix relative to both FFAR1 and the rest of Gq (Fig. 3c, Supplementary Fig. 13). Because the α5 helix extends from FFAR1 to the nucleotide-binding site in Gq, such a change could potentiate nucleotide exchange.

### Experimental validation of non-canonical activation mechanism

To validate our molecular mechanism for ICL2-mediated signaling, we conducted several in vitro experiments. First, a key component of our proposed mechanism is that AP8's carboxylate group forms polar interactions that hold ICL2 in a conformation favorable for G protein signaling (Fig. 2). We reasoned that a mutation to FFAR1 that introduces a carboxylate in a similar position could increase constitutive activity of the receptor, effectively mimicking AP8's interactions. We chose to mutate glycine 103 (3.49 in the Ballesteros−Weinstein numbering scheme), which is directly adjacent to the AP8 carboxylate binding site, to glutamate. Simulations of the G3.49E mutant showed the carboxylate sidechain of E3.49 indeed often forms the same interactions as the AP8 carboxylate (Fig. 4c). IP1 accumulation assays in HEK293 cells expressing FFAR1 were used as a measure of Gq-mediated signaling. The G3.49E mutation significantly increased basal activity (normalized for receptor surface expression) in agreement with the proposed mechanism (Fig. 4d, Supplementary Table 1)[36]. By contrast, mutating G3.49 to residues that could not form these hydrogen bonds −including aspartate, whose carboxylate does not extend far enough, as well as leucine and alanine−decreased basal activity (Supplementary Table 1, Supplementary Fig. 14c).

Our computationally derived mechanism elegantly explains this trend, which is otherwise surprising when compared to the behavior of other class A GPCRs. Most class A GPCRs have a carboxylate residue (aspartate or glutamate) at position 3.49, which stabilizes the inactive state; mutating it to a neutral residue typically increases basal activity[37,38]. Our finding that FFAR1's basal activity instead increases when one replaces a neutral residue at this position with glutamate supports our proposed mechanism and underscores the significance of the unique polar network we observed at FFAR1.

Second, we observed that, whereas AP8 increases the fraction of time that ICL2 spends in the PR state in simulation of the wild-type FFAR1, AP8 has the opposite effect at the G3.49E mutant (Fig. 4c). In simulations of the mutant, AP8's carboxylate tends to be pushed outward, away from the glutamate, disrupting the hydrogen bond network that otherwise stabilizes ICL2 in the PR state (Fig. 4a). These computational results suggest, surprisingly, that AP8 should act as an *inverse agonist* at the G3.49E mutant receptor despite acting as an agonist at the wild-type receptor. Indeed, our in vitro experiments showed that AP8 acts as an inverse agonist at the G3.49E receptor: AP8 lowered the IP1 accumulation levels to 40 ± 6% of basal activity. Similarly, the G3.49E mutation converts AP3−a close analog of AP8 that binds at the same membrane-facing site−from an agonist to an inverse agonist (Supplementary Fig. 14a). In contrast, the G3.49E mutation does not alter the efficacy of orthosteric agonist MK-8666, which remains an agonist at the mutant receptor (Fig. 4b, Supplementary Supplementary Table 2).

Interestingly, we observed that an AP8-like ligand (AP3) maintained its positive binding cooperativity with MK-8666 in the G3.49E mutant receptor, despite the fact that AP3 acts as an inverse agonist at this receptor while MK-8666 acts as an agonist (Supplementary Fig. 14a, b). If AP8-like ligands acted by directly stabilizing an active or inactive conformation of the transmembrane helices, as does MK-8666, then an inverse agonist would be predicted to have negative binding cooperativity with an agonist. The observed positive cooperativity supports our conclusion that AP8-like ligands control receptor activation through ICL2

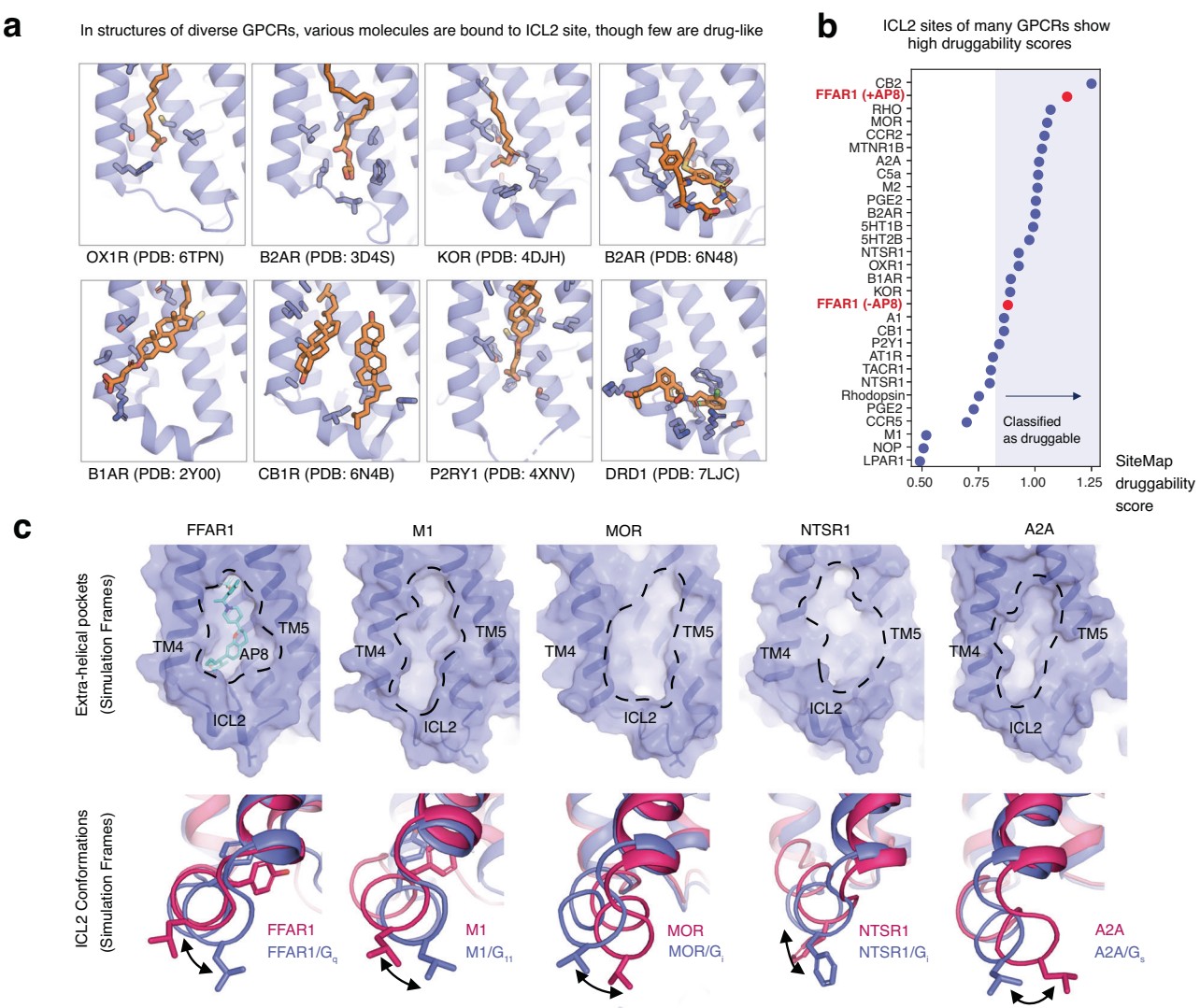

**Fig. 5 | Druggable allosteric pockets exist at the same membrane-facing site in diverse GPCRs. a** Curated structures from the Protein Data Bank (PDB), with non-protein molecules bound to the ICL2 site. Receptors are shown in purple with ligands and other molecules in orange sticks. **b** Using Schrodinger's SiteMap software, we scored the druggability of ICL2 allosteric sites from 29 class A GPCRs, selected for diversity and a well-resolved ICL2 region. Representative static structures from the PDB were used. Druggable classification is determined from previously reported literature benchmarks. **c** Selected frames from simulations of a range of different GPCRs show comparable allosteric pockets to the AP8 binding pocket. All receptors are shown from the same view angle with the putative pockets highlighted by dashed lines. Below, selected simulation frames show changes in the ICL2 helix angle and orientation upon formation of receptor-G protein complex, suggesting functional relevance. Purple structures were selected from simulations of receptor-G protein complexes, red structures were selected from simulations of receptors only.

instead, independent of other effects on the receptor that may contribute to ligand binding cooperativity. We note that in simulations with MK-8666 bound, AP8 has a stabilizing effect on the extracellular end of TM4 that contacts both ligands (Supplementary Fig. 2a), and indeed the presence of AP8 leads to reduced mobility of MK-8666 consistent with previous studies[39]. This effect may underlie the observed binding cooperativity between the two ligands.

As an additional test of our conclusion that AP8 acts not by stabilizing a helical ICL2 over a disordered ICL2 but by stabilizing one helical ICL2 conformation over another, we made a helix-destabilizing mutation of alanine to glycine at position 116, a residue located in the ICL2 helix but not directly contacting AP8. This mutation did not reduce the activity of the AP8-bound receptor, increasing it instead (i.e., Emax increases; see Supplementary Table 1). This supports our simulation findings that stabilizing a folded-over disordered ICL2 helix is not crucial to AP8's agonism.

## Applicability of non-canonical activation mechanism at other GPCRs

Might certain ligands stimulate GPCRs other than FFAR1 via a similar non-canonical activation mechanism? To address this question, we first examined experimentally determined structures of other class A GPCRs. These structures indicate that, in many receptors, diverse molecules bind at a site corresponding to the binding site of AP8, above ICL2 and within the groove formed by TM3/4/5, which we term the "ICL2 site" (Fig. 5a). Most are lipids or sterols, but two are drug-like small molecules: compound 6 at the ß2-adrenergic receptor (B2AR) and mevidalen (LY3154207) at the D1 dopamine receptor (D1R)[28,40]. Interestingly, both compound 6 and mevidalen act not only as positive allosteric modulators for orthosteric agonists but also as weak agonists in their own right[27,40,41].

Second, we analyzed the "druggability" of the ICL2 site at a diverse set of GPCRs—that is, how well the site might bind drug-like small molecules. We analyzed structures of 29 class A GPCRs selected for

diversity and a well-resolved ICL2 region (Fig. 5b). None of these structures had a drug-like ligand bound in the prospective allosteric site. We also included the FFAR1 structures with and without AP8 as references. Schrodinger's SiteMap software was used to score druggability, based on the volume, curvature, and polarity of each potential binding site. The majority of the 29 GPCRs had ICL2 sites with scores in the range empirically predicted to be druggable[42] (Fig. 5b). Most had druggability scores higher than the AP8-free FFAR1 structure. Despite the absence of a drug-like ligand in the ICL2 site of these structures, several—such as the mu-opioid receptor (MOR) and CB2 cannabinoid receptor (CB2R)—had druggability scores comparable to the AP8-bound FFAR1 structure.

These observations suggest that drug-like ligands could be designed to bind specifically to the ICL2 site in many GPCRs, but would such ligands be able to modulate signaling? We analyzed this site and ICL2 conformation in simulations of muscarinic (M2), mu-opioid (MOR), neurotensin (NTSR1), and adenosine (A2A) receptors. These are key targets for treatment of pain, addiction, Parkinson's disease, Alzheimer's, schizophrenia, and cancer[3,43–45]. We found that clear pockets within the cleft of TM4 and TM5, comparable to the AP8 pocket, form in each of these GPCRs (Fig. 5c). As at FFAR1, we also found that ICL2 helix angle and orientation differ between simulations of each receptor with and without a G protein bound (Fig. 5c, Supplementary Fig. 15). Ligands at these sites could thus control signaling by stabilizing particular ICL2 conformations and consequently affecting the receptor interaction with G proteins or other signaling partners, such as arrestins.

## Discussion

Our results demonstrate that certain ligands stimulate GPCR signaling via a mechanism fundamentally different from the canonical activation mechanism described extensively in previous work[7]. In the canonical activation mechanism, which undoubtedly underlies the action of the vast majority of GPCR agonists, the binding of an agonist leads to the rearrangement of key transmembrane helices and conserved switches within the receptor core. We find, however, that subtle changes in intracellular loop conformation alone can significantly affect G-protein binding and signaling, and that an allosteric agonist can trigger these changes without first affecting the transmembrane helices. The TM helices (including TM6) are then stabilized in their "active" conformations only as a result of G protein binding. This mechanism is supported by multiple lines of evidence, including structural analyses, extensive MD simulations, mutations that increase constitutive activity in cellular assays, and mutations that convert the ligand from an agonist to an inverse agonist. Although our study focused on FFAR1, analysis of other structures and simulations suggests that one could design allosteric ligands that act via this non-canonical activation mechanism at many important GPCR drug targets. Notably, certain agonists could trigger activation of a receptor by the non-canonical activation mechanism and other agonists could trigger activation of the same receptor via the canonical mechanism, as appears to be the case for FFAR1.

This non-canonical activation mechanism differs substantially from the mechanisms of action previously proposed for other GPCR ligands that bind near ICL2, such as compound 6 and mevidalen. These ligands were proposed to function by causing ICL2 to form a helix, thus triggering a rearrangement of the transmembrane helices connected to it, leading to the same rearrangement of transmembrane helices that characterizes the canonical activation mechanism. By contrast, the activation mechanism we observe does not require ligand-driven rearrangement of transmembrane helices at all. Even in the absence of an agonist, the transmembrane helices are in a conformational equilibrium, occasionally adopting a conformation with an open G-protein binding pocket[46]. When the G protein binds, it will stabilize the transmembrane helices in such a conformation (i.e., the standard "active"

conformation), but binding of AP8 favors G-protein binding by causing an independent, local rearrangement of an intracellular loop.

Our finding that ligands can activate GPCRs by causing intracellular loop rearrangement instead of transmembrane helix motion suggests rich possibilities for GPCR-targeted drug discovery. Most immediately, it provides new avenues for the rational design of agonists or inverse agonists. Ligands might be designed to target the ICL2 site—or other intracellular allosteric sites—and stabilize intracellular loops in a conformation that favors or disfavors G protein stimulation. Such an approach provides opportunities to achieve selectivity between GPCRs with nearly identical orthosteric sites and, of course, to design drugs that do not compete with endogenous ligands and most existing drugs for the orthosteric binding site. We note that the extent to which this non-canonical mechanism can contribute to receptor signaling may depend on factors unique to each receptor, such as the baseline stability of the receptor–G protein complex and the receptor's level of basal activity.

Our results also suggest opportunities for the design of biased drugs that cause GPCRs to favor or avoid stimulation of arrestins relative to G proteins, which could lead to more effective treatments for diverse diseases[47,48]. Previous studies of other GPCRs indicate that ICL2 plays an important role in triggering arrestin activation[49] and that ICL2 can adopt different conformations when binding to arrestins vs. G proteins[50,51]. Ligands that stabilize different ICL2 conformations might thus differentially favor arrestin signaling relative to G protein signaling[52]. Indeed, in functional studies of the 5HT2A serotonin receptor, mutations of ICL2 residues have been shown to dramatically alter arrestin binding relative to G protein binding[53]. Future work might explore how various ligands that bind at the ICL2 site affect arrestin signaling.

Might some endogenous ligands act via a similar non-canonical activation mechanism? Lipids and sterols, being membrane-soluble, are well positioned to interact with the membrane-facing pocket where AP8 binds. Indeed, most of the ligands positioned in this ICL2 site in previously determined GPCR structures are lipids and sterols. It has also been proposed that endogenous fatty acids interact with the ICL2 site in FFAR1, though no fatty acids were observed in this site in recent structures and binding affinity may be weak[16,17,29,54]. To further investigate this possibility, we conducted several exploratory simulations of FFAR1 in which alpha-linolenic acid (ALA)—an endogenous agonist of FFAR1—was modeled into the AP8 binding site (Supplementary Fig 16). In these simulations, ALA favored the same ICL2 conformation as AP8 (Supplementary Fig 16). This leads us to speculate that certain endogenous agonists might act in part via this non-canonical mechanism. Further work will be necessary to test this hypothesis.

## Methods

### System setup for MD simulations

29 simulation conditions with 185 individual simulations were investigated as listed in Supplementary Table 4. For simulation conditions 1-16, the AP8-bound crystal structure of FFAR1 (PDB 5TZY) was used as the starting point. The structure for these simulations was prepared by first removing the co-crystallized T4 lysozyme. Prime (Schrodinger, v2019-1) was used to model in missing side chains and missing extracellular and intracellular loops. The thermostabilizing point mutations were maintained unless specified. For conditions 10-16, 20, and 21 mutations were introduced. Sidechains were mutated using Maestro (Schrodinger, v2019-1), and Maestro's rotamer library was used to select an initial rotamer that best-minimized clashes. For conditions 6-7, ICL2 was first removed and remodeled using an ICL2 segment from condition 2, where ICL2 had adopted the NR state. For condition 17, the AP8-absent crystal structure of FFAR1 (PDB 5TZR) was used and prepared similarly to above. For conditions 18–21, we used homology modeling in Prime to build an active-state model of FFAR1 in a complex

with heterotrimeric Gq. Our active-state model is very similar (RMSD 0.4 Å) to the recently published GPR40 active structure around the AP8 binding site (PDB 8EIT). For modeling the active FFAR1 receptor, we used a composite modeling approach. For TM1-4 and connecting loops, with the exception of the DRY motif on TM3, we used the FFAR1 structure 5TZY as a template. This allowed the preservation of the AP8 binding site. TM5–7 and the DRY motif used B2AR (PDB 3SN6) as a template. The alignment of FFAR1 and B2AR sequences was automatically generated and then corrected to ensure that all helices were accurately modeled as per the 5TZY structure. For modeling Gq, we used templates from a related GPCR G11 complex (PDB:6OIJ) and GPCR Go complex (PDB:6G79). All template structures were first aligned to TM1–j4 of the receptor. Palmitoylations were added to the N-terminus of Gq[55]. Both the active receptor and Gq model were constructed simultaneously in a single Prime model to ensure a clean interface.

For all FFAR1 simulations, interior waters were added from the higher-resolution FFAR1 crystal structure 4PHU. AP8's carboxylate was left deprotonated (except in condition 5), in accordance with its low pKa, solvent accessibility, and the necessity to form hydrogen bonds with surrounding hydrogen bond acceptors. The prepared protein structure was aligned on the transmembrane helices to the Orientation of Proteins in the Membranes (OPM) structure of PDB entry 4PHU[56]. Parameters for AP8, MK-8666, and ALA were generated using the CHARMM General Force Field (CGenFF) with the ParamChem server (v1)[57]. Across all multi-microsecond simulations, the ligands remained stably bound within the binding pocket and formed persistent contacts with surrounding residues.

For conditions 22-29, the PDB structure described in Supplementary Table 4 was downloaded and prepared using the relevant protocols described above.

For all simulations, hydrogen atoms were added, and protein chain termini were capped with neutral acetyl and methylamide groups. PropKa (Schrodinger, v2019-1) was used to determine the dominant protonation state of all titratable residues at pH 7[58]. The Dowser program (1999 release) was used to hydrate any additional pockets within and around the GPCR. Then the receptor was inserted into a pre-equilibrated palmitoyl-oleoyl-phosphatidylcholine (POPC) bilayer using Dabble (v2.7.6)[59,60]. Sodium and chloride ions were added to neutralize each system at a concentration of 150 mM. Approximate system dimensions were 80 Å x 90 Å x 85 Å for receptor-only simulations, and 120 Å x 120 Å x 140 Å for receptor–G protein complexes. We used the CHARMM36 parameter set for protein molecules, lipids, and ions, and the CHARMM TIP3P water model for waters[61,62].

## Simulation protocols

All simulations were run on a single Graphical Processing Unit (GPU) using the Amber18 Compute Unified Device Architecture (CUDA) version of particle-mesh Ewald molecular dynamics (PMEMD)[63]. For each independent simulation, the system was minimized with 500 steps of steepest descent followed by 500 steps of conjugate gradient descent three times. 10 and 5 kcal mol$^{-1}$ Å$^{-2}$ harmonic restraints were used on the protein, lipid, and ligand atoms for the first and second minimization, respectively. 1 kcal mol$^{-1}$ Å$^{-2}$ harmonic restraints were used on the protein and ligand atoms for the final minimization. The systems were then heated over 12.5 ps from 0 K to 100 K in the NVT ensemble using a Langevin thermostat with harmonic restraints of 10.0 kcal·mol$^{-1}$·Å$^{-2}$ on the non-hydrogen atoms of the lipids, protein, and ligand. Initial velocities were sampled from a Boltzmann distribution. The systems were then heated to 310 K over 125 ps in the NPT ensemble. Equilibration was performed at 310 K and 1 bar in the NPT ensemble, with harmonic restraints on the protein and ligand non-hydrogen atoms tapered off by 1.0 kcal·mol$^{-1}$·Å$^{-2}$ starting at 5.0 kcal·mol$^{-1}$·Å$^{-2}$ in a stepwise manner every 2 ns for 10 ns, and finally by 0.1 kcal·mol$^{-1}$·Å$^{-2}$ every 2 ns for an additional 18 ns. All restraints

were completely removed during production simulation. For standard molecular dynamics (all conditions except 8 and 9), production simulations were performed at 310 K and 1 bar in the NPT ensemble using the Langevin thermostat and Monte Carlo barostat. The simulations were performed using a timestep of 4.0 fs while employing hydrogen mass repartitioning[64]. Bond lengths were constrained using SHAKE. Non-bonded interactions were cut off at 9.0 Å, and long-range electrostatic interactions were calculated using the particle-mesh Ewald (PME) method with an Ewald coefficient (β) of approximately 0.31 Å and B-spline interpolation of order 4. The PME grid size was chosen such that the width of a grid cell was approximately 1 Å. Snapshots from each trajectory were saved every 200 ps during the production phase of each simulation. The AmberTools18 CPPTRAJ package (v16)[65] was used to reimage trajectories, while Visual Molecular Dynamics (VMD, v1.9.3),[66] PyMol (Schrodinger, v2.3.2), Matplotlib Python package (3.6.2) were used for visualization and analysis.

For simulation conditions 8 and 9, we employed adaptively biased molecular dynamics (ABMD) in Amber18[30,63]. Specifically, we used flooding mode, with a flooding timescale of 200 picoseconds and a monitor frequency of 5000 picoseconds. All other production settings were the same as those previously described. We defined a multi-RMSD collective variable using the backbone nonhydrogen atoms of ICL2 residues 111-118. For sampling free energy along this collective variable, we specified a resolution of 0.2, a minimum of 0, and a maximum of 5.5. The free energy along this coordinate was then collected in a separate output file.

## ICL2 conformational analysis

ICL2 helicity was determined by measuring the fraction of backbone hydrogen bonds between residues $i$ and $i + 4$ on ICL2 combined with an additional RMSD (root-mean-square deviation) cutoff. Here, ICL2 was defined as FFAR1 residues 110 to 118. Hydrogen bond detection was performed with standard geometric criteria using the getcontacts software tool (https://getcontacts.github.io/). The RMSD of the ICL2 segment was calculated on ICL2 backbone atoms after aligning this selection to the starting frame, where ICL2 is helical. If 3 or more backbone hydrogen bonds ($i, i + 4$ only) were present in the frame or the RMSD of ICL2 backbone atoms was <2 angstroms, the ICL2 conformation of that frame was classified as helical. In the initial helix, only 4 backbone hydrogen bonds are present and we found the cutoff of 3 helical hbonds was sufficient to accurately capture the state while allowing some flexibility.

We then created two metrics to describe the different ICL2 helical conformations. First, we calculated the angle (about the helical axis) of ICL2 relative to the rest of the receptor. The receptor was aligned on TM1,2,3, and 4. We found that this placed the helical axis of ICL2 approximately perpendicular to the $z,y$ plane. Then, a vector was drawn between atoms 114:OH and 112:CG in the z,y plane. The beginning of this vector was placed at a hypothetical origin, and then the ICL2 angle was calculated as on a typical unit circle. For displayed traces of this angle, we then set the initial angle in the crystal structure as 0 degrees, to provide a convenient point of reference. As a second metric, we calculated the distance between Y114 and TM2 (using the Y114:OH atom to T39:CB atom). This allowed us to detect whether Y114 had moved into or away from the helical bundle.

To calculate the fraction of time spent in the PR or NR state, we developed a set of simple thresholds to assign the state. These thresholds were based on the ICL2 angle ($a$) and the Y114 to TM2 distance ($d$) in order to create well-separated clusters of states. The PR state was assigned if ICL2 was helical, $45 < a < 120$, and $10 - a/20 < d < 13$. The rotated state was assigned if ICL2 was helical but not in the crystal state. The overall results were robust to small changes in these thresholds.

For the more generalized ICL2 analysis shown in Supplementary Fig. 15, we used the machine learning library sklearn to perform

principal component analysis of the phi, psi backbone dihedrals of ICL2 residues 3.55 to 4.39. We only used frames where ICL2 was helical by applying the RMSD cutoff described previously. For each receptor, we calculated new principal components using merged data across available conditions and simulation trials for that receptor.

## Additional simulation analysis

To quantify the vertical shift of TM5 relative to TM4, shown in Fig. 1a and Supplementary Fig. 2, we projected the position of a Cα atom on TM5 (residue 190), onto the line connecting the Cα atoms of TM4 residues 130 and 141. We then report the position of the projected point on that line, using the convention that the AP8-free crystal structure is at 0 and positive values indicate an upward shift of TM5 toward the extracellular side of the receptor.

To quantify the frequency of hydrogen bonds, shown in Fig. 2b, we used the getcontacts software tool. Hydrogen bond detection was performed with standard geometric criterion. The frequency of hydrogen bonding interactions for an atom pair was calculated by taking the number of simulation frames where a hydrogen bond or water-mediated hydrogen bond between specified atoms was detected and dividing by the total number of simulation frames. The average frequency of each interaction over the ten simulations for each condition was calculated.

To investigate the effect of AP8 on G-protein conformation, we calculated the displacement of Gα helix 5 relative to the receptor for each condition (shown in Fig. 3c). After aligning receptor TM helices 1–4 to the starting frame, we calculated the RMSD of Gqα residues 519 to 530 for each frame. To measure G-protein internal conformation, we analyzed β6-α5 loop (Supplementary Fig. 13). We calculated the end-to-end distance of the loop using residue 504 Cα to residue 508 Cα. The average distance over the five simulations for each condition was calculated. We also calculated the average standard deviation of this quantity.

## IP accumulation assay

Human GPR40 WT and mutants were transiently expressed in HEK293 cells (CRL-1573, purchased from American Type Culture Collection and mycoplasma tested). HEK293 cells were grown in Dulbecco's Modified Eagle Medium (DMEM) containing 10% Fetal Bovine Serum (FBS), 1% penicillin, and streptomycin (Life Technologies). 40,000 cells per 100 µl per well were seeded in a 96-well poly-d lysine-coated plate. Transfection complexes were prepared by adding 5 ug of plasmid DNA (pcDNA 3.4 TOPO) to 300 µl of optiMEM (Life Technologies), and 18 µl of Fugene HD (Promega) to 300 µl of optiMEM. These two solutions were mixed, and incubated at room temperature for 20 min, and then 10 µl of this solution was added per well to cells in the 96-well cell culture plate (84 ng plasmid DNA per well). 24 hours post-transfection, the media was changed to optiMEM. 48 hours post-transfection, IP1 accumulation assay was performed. On the day of the experiment, the growth media was removed from the assay plate and 40 µl of IP1 stimulation buffer (Cis Bio IP-one Tb HTRF kit) supplemented with 50 mM LiCl added to each well. Test compounds dissolved in DMSO were serially diluted in half-log increments, starting from 1 mM as 100X, diluted to 10X in stimulation buffer before adding 5 µl per well (final starting concentration 10 µM). Plates were then incubated for 60 min at 37 °C. 50 µl of Lysis buffer per well was added to each plate and incubated for 60 min at room temperature. 10 µl of detection buffer (prepared as described in the Tb HTRF kit) is added to each well. The plates are then incubated an additional 1 hr and 30 min at room temperature. After the final incubation, the plates were read in a Perkin Elmer Envision with a method designed for HTRF assays (320 nm excitation, dual emission 615 and 655 nm). For each assay, a standard curve plate in which IP1 is titrated is also included. All fluorescent readings (using the 655/615 nm ratio) are back-calculated to a concentration of IP1 using the IP1 standard curve. The percent activity at

each concentration of the test compound is normalized using the basal activity of WT GPR40 determined in GPR40 WT wells that contained DMSO. The % activation is then plotted versus the concentration of the test compound and the dose-response curve fitted to a standard 4-parameter nonlinear regression model using a GraphPad Prism 7. Maximal % activation and EC50 are then determined for each test compound.

## Plasmid construction

WT GPR40 was cloned in a pcDNA3.4 TOPO TA vector with a N-terminal Flag tag. GPR40 mutants were generated using site-directed mutagenesis at GENEWIZ.

## Whole-cell ligand binding assay

Human GPR40 WT and mutants were transiently expressed in HEK293 cells (purchased from American Type Culture Collection (ATCC) and mycoplasma tested). HEK293 cells were grown in DMEM containing 10% FBS (FBS), 1% penicillin, and streptomycin (Life Technologies). 10 million cells were seeded in a 10 cm cell culture dish and were transiently transfected with 17 µg of plasmid cDNA (pcDNA 3.4 TOPO) and 53 µl Fugene HD (Promega) in 10 ml of HEK293 media. Following 48 hr incubation at 37 °C and 5% CO2, the transiently transfected HEK293 cells were harvested using dissociation buffer TrypLE (Thermo Fisher Scientific) and binding assay buffer. The cells were pelleted (1000 r.p.m. for 5 min) and resuspended in binding assay buffer (50 mM Tris HCl, 5 mM MgCl2, 1 mM CaCl2, 100 mM NaCl and 0.1% fatty acid-free BSA, Sigma, pH 7.4). For the assay, [3H]-labeled P4, AP8 or AP3 (two-fold serially diluted in binding buffer with a top working concentration of 500 nM), and 60,000 cells were added to a 2-ml 96-well master block plate (Greiner Bio-One) in a total volume of 125 µl. The plate was incubated for 4 h at room temperature and the assay was then harvested onto a GF/C filter plate (PE) that had been presoaked in 0.3% polyethyleneimine (Sigma) using a Packard Filter Mate Harvester. The plate was washed 4× with 1 ml cold wash buffer (50 mM Tris HCl, 5 mM MgCl2, 500 mM NaCl, 2 mM EDTA and 0.05% Tween 20), dried for 1 h at 40 °C and then 50 µl of MicroScint 20 (PE) was added to each well. Plates were then read on a TopCount scintillation counter. Nonspecific binding was determined by the addition of 50 times of cold P4, AP8 or AP4 to control wells and was subtracted from all wells of total binding, in which DMSO was added instead of cold compounds. Kd values were determined by a standard one-site specific binding model.

## Expression analyses by flow cytometry

HEK293 cells transfected with WT or mutant GPR40 plasmids for IP1 assay and or Binding Assay were harvested with TrypLE (Thermo Fisher Scientific), resuspended in optiMEM (Life Technologies) with 5% heat inactivated Fetal bovine serum (HI-FBS) (Gibco). 1 million cells per 100 µl per sample were stained with anti-Flag antibody (EPR20018-251, Abcam) (1:100) at Room temperature for 2 hours. Cells were washed 3 times with optiMEM+5% HI-FBS, then stained with a secondary antibody Anti Rabbit-Alexa 488 (#4412, Cell Signaling Technology) (1:200) for 1 hr at room temperature. Cells were then washed again, resuspended in optiMEM+5%HI-FBS, and read on an Accuri C6 flow cytometer. Median Fluorescence intensities of Alexa-488 of 20,000 cells per sample were collected and used to calculate % of WT where WT intensities were normalized to 100%.

## Reporting summary

Further information on research design is available in the Nature Portfolio Reporting Summary linked to this article.

# Data availability

Simulation trajectories and ligand parameters generated in this study are available at https://doi.org/10.5281/zenodo.13864831. Additional data supporting the findings of this work are included in

Supplementary Information and Source Data file. Structural models used in this study were accessed from the Protein Data Bank under accession codes 5TZR (GPR40, AP8-bound) and 5TZY (GPR40). Source data are provided with this paper.

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

## Acknowledgements
Funding was provided by National Institutes of Health (NIH) grant R01GM127359 (R.O.D.), a National Science Foundation Graduate Research Fellowship (A.S.P.), the Stanford ChEM-H Chemistry/Biology Interface Predoctoral Training Program under NIH Award T32GM120007 (A.S.P.), and a Stanford Graduate Fellowship (J.M.P.). We thank Helen Yu for assistance with druggability calculations.

## Author contributions
A.S.P. and J.M.P. performed molecular dynamics simulations. A.K., S.S., S.M.S., J.D.S., and J.L. performed mutagenesis, ligand binding, and activity experiments. A.S.P., J.M.P, and N.R.L. analyzed simulations and performed structural analyses. R.O.D., J.M.J., and A.B.W. supervised the project. A.S.P., A.K., and R.O.D. wrote the paper with input from all authors.

## Competing interests
A.K., S.S., J.D.S., J.L., S.M.S., J.M.J., and A.M.W. are current or past employees of Merck Research Laboratories. R.O.D. holds equity in Septerna Inc. The remaining authors declare no competing interests.
