## [Peer Review file · Nature Communications]

A non-canonical mechanism of GPCR activation

Corresponding Author: Professor Ron Dror

This manuscript has been previously reviewed at another journal. This document only contains reviewer comments, rebuttal and decision letters for versions considered at Nature Communications.

Version 0:

Reviewer comments:

Reviewer #1

(Remarks to the Author)

The authors have partially addressed the previous concerns. There are no additional comments at this time. However, the authors should cite and discuss a previous publication on similar studies, specifically MD simulations on FFAR1 with MK-8666 and AP8 (PMID: 34377364).

Reviewer #2

(Remarks to the Author)

The authors analyze primarily with long MD calculations how the AP8 compound binds a membrane proximal site of a partially activated GPCR to fully activate it. The authors have carefully addressed my concerns as far as possible. The intracellular loop ICL2 was known to be functionally relevant but not the mechanism. Here, the authors reveal a plausible mechanism how this can modulate the effect of binding membrane proximal small molecule ligands.

Fig. 5 presents potentially valuable predictions on how a some GPCRs could be target at the membrane-proximal site, which could be tested experimentally.

This is an excellent manuscript and will enrich the understanding of some GPCR function.

Reviewer #4

(Remarks to the Author)

The manuscript describes extensive molecular dynamics simulations of FFAR1 bound to orthosteric and/or allosteric ligands to investigate the allosteric activation mechanism. The results unveil a distinct activation mechanism by the allosteric agonist AP8, supported by site-directed mutagenesis experiments. Notably, the simulations accurately predicted mutagenesis outcomes, which makes the work even more convincing. The study represents a rigorous experimental design including ample control simulations and experiments. More interestingly, the authors propose that similar allosteric sites may be present in GPCRs, opening potential new drug design opportunities.

My main concern is about AP8's lack of impact on the transmembrane helices as a distinct feature. The authors refer to the common canonical mechanism of class A GPCR activation in which agonists cause the opening of the intracellular transducer-binding pocket. However, here Fig S1 shows that the orthosteric agonist MK-8666 did not induce such an opening either. This raises 2 questions:

1) Microsecond-timescale MD simulations without enhanced sampling are often insufficient for capturing such intracellular opening. It is uncertain whether the lack of intracellular opening by either ligands is intrinsic to the ligands or due to insufficient MD sampling. This does not invalidate the main finding of the study about AP8's distinct effects on FFAR1. However, its lack of impact on the transmembrane helices should be interpreted and discussed with caution.

2) Is it possible that (orthosteric or allosteric) agonists alone cannot trigger the intracellular opening without the transducers. This has been demonstrated in the mu-opioid receptor, for instance: Sounier et al. Nature 2015 and Zhao et al. Nature 2024.

Indeed, the notion that class A GPCR agonists activate the receptors by opening the intracellular pocket is mostly deduced from high-resolution structures bound with transducers or mimetic nanobodies. Therefore, this point should be clarified in the discussion when it comes to the statements on AP8's mechanism of action, e.g. "... again suggesting that AP8 does not act by directly stabilizing the active-state conformation of the TM helices.". In my opinion, the lack of impact on the transmembrane helices may or may not be due to insufficient MD sampling. This cannot be concluded based on the data in the current study.

Minor comments

- Fig. S1. While the plots clearly illustrates the differences between MK-8666 and AP8, it would be helpful to include 3D illustrations of the measuring points on each TM to help readers appreciate the conformational changes without looking up the PDBs.

- Why did the simulations have different lengths, and why did some have 10 trials others had 5?

We are pleased that the reviewers of the original manuscript were satisfied with our revision, and we appreciate reviewer 2's statement that "This is an excellent manuscript." The only further request from these reviewers was that we add one citation (to PMID: 34377364). We have added this citation on p. 9 of our second revision.

We thank reviewer 4 for agreeing to review our revised manuscript despite not having reviewed the original. In our second revision, we have clarified several points, as requested by the reviewer. Below, we include the reviewer's comments in italicized, indented text, followed by our responses.

Reviewer #4:

The manuscript describes extensive molecular dynamics simulations of FFAR1 bound to orthosteric and/or allosteric ligands to investigate the allosteric activation mechanism. The results unveil a distinct activation mechanism by the allosteric agonist AP8, supported by site-directed mutagenesis experiments. Notably, the simulations accurately predicted mutagenesis outcomes, which makes the work even more convincing. The study represents a rigorous experimental design including ample control simulations and experiments. More interestingly, the authors propose that similar allosteric sites may be present in GPCRs, opening potential new drug design opportunities.

We appreciate the reviewer's enthusiastic comments.

My main concern is about AP8's lack of impact on the transmembrane helices as a distinct feature. The authors refer to the common canonical mechanism of class A GPCR activation in which agonists cause the opening of the intracellular transducer-binding pocket. However, here Fig S1 shows that the orthosteric agonist MK-8666 did not induce such an opening either. This raises 2 questions:

1) Microsecond-timescale MD simulations without enhanced sampling are often insufficient for capturing such intracellular opening. It is uncertain whether the lack of intracellular opening by either ligands is intrinsic to the ligands or due to insufficient MD sampling. This does not invalidate the main finding of the study about AP8's distinct effects on FFAR1. However, its lack of impact on the transmembrane helices should be interpreted and discussed with caution.

2) Is it possible that (orthosteric or allosteric) agonists alone cannot trigger the intracellular opening without the transducers. This has been demonstrated in the mu-opioid receptor, for instance: Sounier et al. Nature 2015 and Zhao et al. Nature 2024. Indeed, the notion that class A GPCR agonists activate the receptors by opening the intracellular pocket is mostly deduced from high-resolution structures bound with transducers or mimetic nanobodies. Therefore, this point should be clarified in the discussion when it comes to the statements on AP8's mechanism of action, e.g. "... again suggesting that AP8 does

not act by directly stabilizing the active-state conformation of the TM helices." In my opinion, the lack of impact on the transmembrane helices may or may not be due to insufficient MD sampling. This cannot be concluded based on the data in the current study.

We agree that one would not expect to see intracellular opening of the transducer-binding pocket in microsecond-timescale simulations, and we have clarified this point in our revised manuscript (p. 4-5).

We have also clarified the evidence for AP8's lack of impact on transmembrane helices, as described below.

First, our simulations of AP8-bound and AP8-free FFAR1 started with the transmembrane (TM) helices in substantially different conformations, reflecting the fact that in the AP8-bound and AP8-free FFAR1 crystal structures, certain TM helices adopt different conformations. In our simulations, the TM helices converge to the same conformation with and without AP8 bound. We have added a new figure (Supp. Fig. 3) to illustrate this in more detail. These simulations suggest that the differences in the crystal structures are explained by crystal packing differences (Supp. Fig. 2b, 2c) rather than reflecting the AP8-induced activation mechanism as had previously been suspected. If AP8 in fact had some substantial direct effect on TM helix position, we would expect that simulations started from these two different conformations would likely not converge to quite the same conformation (even though both the AP8-bound and AP-free simulations remain in inactive conformations and do not capture intracellular opening).

By contrast, the presence or absence of the orthosteric partial agonist MK-8666 has significant effects on TM helix positions in *both* crystal structures and simulations. Comparing simulations with and without MK-8666, we see persistent differences (up to 3 Å) in the distances between certain regions of transmembrane helices including TM5 and TM6 (Supp. Fig. 1c)—even though these simulations do not capture intracellular opening. We observed these differences despite the fact that MK-8666 is only a partial agonist, whereas AP8 is a full agonist.

Second, taking advantage of the recently published active-state FFAR1 structure (the FFAR1-G protein complex), we compared the conformation of the TM helix residues surrounding AP8's binding site in the active and inactive state. The conformation of both the backbone and sidechains of the TM helix residues around the AP8 binding site closely matched (deviation of 0.6 Å, Supp. Fig. 4a), suggesting that it would be difficult for AP8 to cause activation by influencing the conformation of these residues.

This is in marked contrast to the TM helix residues surrounding the orthosteric binding site, which do undergo significant conformational changes between the active and inactive states. These changes include rearrangements of sidechain rotamers and polar interactions with residues on TM3, TM6, and TM7, as illustrated in a new figure we have added (Supp. Fig. 5).

Third, we performed simulations of AP8 bound to both the active state and inactive state. We quantitatively analyzed AP8's interactions with each residue surrounding the binding site (Supp. Fig. 4b). If AP8 caused receptor activation by acting on the TM helices, one would expect its

interactions with TM helix residues to be more favorable in the active state. We found instead that none of AP8's interactions with TM helix residues were significantly more favorable in the active state than in the inactive state. By contrast, multiple residues on ICL2 form more favorable interactions with AP8 in the active state than in the inactive state.

Minor comments

- Fig. S1. While the plots clearly illustrates the differences between MK-8666 and AP8, it would be helpful to include 3D illustrations of the measuring points on each TM to help readers appreciate the conformational changes without looking up the PDBs.

As the reviewer suggested, we have added 3D illustrations to Supp. Fig. 1 to illustrate the measuring points on each TM.

- Why did the simulations have different lengths, and why did some have 10 trials others had 5?

We conducted simulation under 29 different conditions, for a total of 185 individual simulations. Given the finite computational resources available to us and the extensive scope of our investigation, we had to make strategic decisions about resource allocation. For each of the conditions that were most central to our investigation, we performed 10 simulations (trials), each 2 μ s in length. For other conditions, we performed 5 simulations, each 1 μ s in length. In any analysis that required us to compare 1- μ s simulations to 2- μ s simulations, we analyzed only the first microsecond of each simulation, in order to ensure an apples-to-apples comparison.